# Phonons in Short-Period GaN/AlN Superlattices: Group-Theoretical Analysis, *Ab initio* Calculations, and Raman Spectra

**DOI:** 10.3390/nano11020286

**Published:** 2021-01-22

**Authors:** Valery Davydov, Evgenii Roginskii, Yuri Kitaev, Alexander Smirnov, Ilya Eliseyev, Dmitrii Nechaev, Valentin Jmerik, Mikhail Smirnov

**Affiliations:** 1Ioffe Institute, 194021 St. Petersburg, Russia; e.roginskii@mail.ioffe.ru (E.R.); yu.kitaev@mail.ioffe.ru (Y.K.); alex.smirnov@mail.ioffe.ru (A.S.); ilya.eliseyev@mail.ioffe.ru (I.E.); nechayev@mail.ioffe.ru (D.N.); jmerik.pls@mail.ioffe.ru (V.J.); 2Department of Physics, Saint-Petersburg State University, 199034 St. Petersburg, Russia; smirnomb@rambler.ru

**Keywords:** GaN/AlN superlattices, molecular beam epitaxy, group theory analysis, density functional theory, lattice dynamics, Raman spectroscopy

## Abstract

We report the results of experimental and theoretical studies of phonon modes in GaN/AlN superlattices (SLs) with a period of several atomic layers, grown by submonolayer digital plasma-assisted molecular-beam epitaxy, which have a great potential for use in quantum and stress engineering. Using detailed group-theoretical analysis, the genesis of the SL vibrational modes from the modes of bulk AlN and GaN crystals is established. *Ab initio* calculations in the framework of the density functional theory, aimed at studying the phonon states, are performed for SLs with both equal and unequal layer thicknesses. The frequencies of the vibrational modes are calculated, and atomic displacement patterns are obtained. Raman spectra are calculated and compared with the experimental ones. The results of the *ab initio* calculations are in good agreement with the experimental Raman spectra and the results of the group-theoretical analysis. As a result of comprehensive studies, the correlations between the parameters of acoustic and optical phonons and the structure of SLs are obtained. This opens up new possibilities for the analysis of the structural characteristics of short-period GaN/AlN SLs using Raman spectroscopy. The results obtained can be used to optimize the growth technologies aimed to form structurally perfect short-period GaN/AlN SLs.

## 1. Introduction

In modern semiconductor devices, short-period superlattices (SLs) are widely used in the active and buffer regions of various heterostructures and have a great potential in quantum and stress engineering. The SLs based on the (Al, Ga) N bulk crystals can be employed in both the ultraviolet (UV) and infrared (IR) optoelectronics. In the first case, the ability of GaN/AlN SLs with a period of several atomic layers to possess a precisely adjusted effective band gap from 3.4 to 6.1 eV can be used in light-emitting and adsorption layers instead of traditional ternary AlGaN alloys [1,2,3,4,5,6,7]. GaN/Al(Ga)N SLs with a slightly longer period are a promising material system for the implementation of unipolar devices operating in the range from the near IR to the THz one [8,9,10,11]. Moreover, the introduction of different SLs into all heterostructure devices grown on mismatched substrates is very effective for controlling stresses and filtering threading dislocations [12,13,14].

From a fundamental point of view, short-period SLs can be considered as a metamaterial that simultaneously possesses both a number of properties inherent in conventional solid solutions (effective band gap, average lattice parameter) and several completely new properties that are not comprehensively studied at the moment. The effective use of such periodic structures requires a detailed study of their fundamental physical properties, as well as the development of new methods of quantitative diagnostics in order to improve the technology of their growth.

Raman spectroscopy is a recognized tool for the non-destructive study of the phonon spectrum of SLs. Group-theoretical analysis and model methods for studying the phonon spectra form the basis for quantitative analysis of the experimental information. A joint analysis of the theoretical and experimental results makes it possible to establish the microscopic nature of the vibrational states associated with the lines in the Raman spectra, which is the basis for the development of a quantitative technique for assessing important parameters of the studied SLs. The existence of localized and delocalized modes in GaN/AlN SLs at the frequencies different from the frequencies of any mode inherent in the layers composing the SL was theoretically predicted within the framework of the dielectric continuum model [15,16]. It was shown that phonons propagating along the SL growth direction have a localized nature, whereas the phonons propagating in the perpendicular direction are delocalized. At the same time, the microscopic nature of many vibrational states observed in the Raman spectra of SLs has not been sufficiently studied. We propose an approach that includes a comprehensive group-theoretical analysis of the origin of vibrational SL modes from the modes of bulk AlN and GaN crystals, *ab initio* calculations of lattice dynamics, and Raman measurements of the phonon modes in short-period hexagonal GaN/AlN SLs. The results of such an integrated approach can be useful for obtaining new information on the microscopic nature of phonon SL modes and establishing correlations between the structure of the SL and the observed features in the Raman spectra, which can be used to optimize the parameters of the growth process in order to form structurally perfect short-period GaN/AlN SLs.

## 2. Experimental and Calculation Details

The short-period GaN/AlN SLs were grown using plasma-assisted molecular beam epitaxy (PA MBE) setup Compact 21T (Riber, Bezons, France) equipped with a N_2_-plasma source HD-25 (Oxford Appl. Res. Ltd., Oxfordshire, UK), as described in detail in [17]. At first, the growth of AlN templates on annealed and nitridated *c*-sapphire substrates was initiated using migration-enhanced epitaxy of 65-nm-thick nucleation layers at a substrate temperature of 780 °C. Then, upper AlN buffer layers with a thickness of 210 nm were grown using standard PA MBE at the same substrate temperature under slightly Al-rich conditions with the fluxes of Al atoms and plasma-activated nitrogen (N_2_***) of 0.51 and 0.50 ML/s, respectively. After the completion of the buffer layer growth, its surface was exposed to a nitrogen flux to transfer the excess Al with a nominal thickness of ~15 ML to the AlN layer. The SLs *N* × {(GaN)*_m_*/(AlN)*_n_*} were grown with the number of periods *N* = 252–572; the thicknesses of GaN quantum wells (QWs) and AlN barrier layers (BLs) were *m* = 4–9 and *n* = 4–9, respectively, where the thicknesses are given in units of monolayers (MLs) with one-ML-thicknesses of 0.2593 and 0.2491 nm for GaN and AlN, respectively. All SLs were grown at a constant substrate temperature of 690 °C and a N_2_* flux of 0.50 ± 0.005 ML/s with sequential calibrated Ga and Al fluxes with a III/V ratio of ~2 and ~1.1, respectively. To evaporate excess free metal (Ga) atoms from the surface, we used periodic (with the *K* number) growth interruptions with all closed cells for ~60 s after the growth of each *L* = (5–13) SL periods (*N* = *K* × *L*). The substrates were rotated at a constant speed of 30 rpm. The growth rate and surface morphology of the layers and SLs were monitored in situ using home-made laser interferometry (532 nm) and reflected high energy electron diffraction (RHEED) (Staib Instruments GmbH, Langenbach, Germany), respectively. The substrate temperature was measured with an IR-pyrometer (Mikron M680, Mikron Infrared, Inc, Oakland, NJ, USA).

The Raman measurements were performed at room temperature using a T64000 (Horiba Jobin-Yvon, Lille, France) spectrometer equipped with a confocal microscope. The line at 532 nm (2.33 eV) of Nd:YAG laser (Torus, Laser Quantum, Inc., Edinburg, UK) was used as the excitation source. All spectra were measured in backscattering geometry. The scattering geometries are given in Porto’s notation, for example, $z(xx)\bar{z}$. Here, *z* is the direction of the 3-fold optical axis, and *x* and *y* are mutually orthogonal axes, which are arbitrarily oriented in the substrate plane.

The *ab initio* calculations using the plane-wave pseudopotential method were carried out within the framework of density functional theory in the local density approximation (DFT-LDA) as realized in the ABINIT software package [18,19,20]. The norm-conserving pseudopotentials were constructed according to the scheme described in [21]. For Ga, Al, and N atoms, the 3*d*4*s*4*p*, 3*s*3*p*, and 2*s*2*p* electrons were treated as the valence ones. The plain-wave size was controlled by cutoff energy *E_c_* = 45 Ha. The Brillouin zone (BZ) sampling according to the Monkhorst–Pack scheme [22] was chosen as 6 × 6 × 4. The full geometry optimization was performed by varying the parameters of the unit cell and the positions of atoms in it. The convergence tolerance for geometry optimization was selected within 10^−5^ Ha/Bohr and 0.01 GPa in maximal force and stress tensor, correspondingly. The phonon eigenvectors, frequencies, and Raman tensor components were calculated in the Г-point of the BZ within the density functional perturbation theory (DFPT) [23,24]. The Raman spectra were simulated using Lorentzian line shape functions with full width at half maximum of 8 cm^−1^ (unless otherwise specified).

## 3. Results and Discussion

### 3.1. Growth

To understand the growth of GaN/AlN SLs under metal-rich conditions using submonolayer digital PA MBE, we analyzed the desorption kinetics of both Al and Ga atoms using desorption rates equal to zero for the former and 0.22 ML/s for the latter at a substrate temperature of 690 °C, as it has been determined in [17]. The Al-rich conditions used for the growth of AlN layers to precisely control their thickness using a calibrated flux of N_2_* led to the accumulation of Al adatoms due to the lack of their desorption. However, even in the case of the thickest AlN layers in (GaN)_8_/(AlN)_8_ SL, the amount of Al adatoms accumulated during the growth of one AlN layer in the SL did not exceed 0.2 ML. Therefore, although after the completion of AlN growth and switching of the Al/Ga shutters, these excess adatoms are incorporated in the AlN layer instead of the nominal GaN layer due to the higher binding energy for the AlN compound as compared to GaN, the effect of expansion of the AlN layer relative to its nominal thickness can be neglected.

It should also be noted that a small excess of Al flux did not ensure the formation of an Al-bilayer on the AlN surface, which, as is known, leads to two-dimensional (2D) growth of AlN layers using PA MBE [25]. Meanwhile, RHEED exhibited a continuous streaky pattern throughout SL growth (Appendix A), indicating its constant 2D-surface morphology. This fact can be explained by the surfactant effect of Ga adatoms occurring on the SL surface not only during the growth of GaN layers, but also during the subsequent growth of AlN layers. Indeed, the surfactant effect, resulting in the surface smoothing due to an increase in the surface mobility of adatoms, was widely observed for the PA MBE of bulk AlN layers and GaN/AlN multiple QW structures with the addition of non-incorporated Ga adatoms [26,27].

The strongly Ga-enriched growth conditions of the GaN layers led to the very rapid (<2 s) formation of a Ga-bilayer on the surface of these layers, and then excess Ga adatoms accumulated in small nanoclusters, as usually happens in the conventional metal-modulated epitaxy of binary III-Nitrides [17,28,29]. It can be assumed that the Ga nanoclusters, due to the low Ga-desorption rate ${\mathrm{Ga}}^{d}$~0.2 ML/s, supply sufficient Ga adatoms to form a Ga-bilayer on the surface of Al layers throughout their entire growth. A detailed analysis of excess Ga adatoms will be reported elsewhere. Here, the excess of Ga atoms accumulated during the growth of one GaN layer for a time of $t_{\mathrm{GaN}}$ was approximately estimated from the difference in fluxes $\left(\mathrm{Ga}-N_{2}^{*}-{\mathrm{Ga}}^{d}\right)\times t_{\mathrm{GaN}}$. Then we estimated the time required for the complete evaporation of this volume of Ga atoms at the same desorption rate. A greater or equal value of this time found for all SLs compared to the growth time of the AlN layer indicated a sufficient amount of Ga atoms for continuous coverage of the AlN layer surface with a Ga-bilayer.

The above estimates were experimentally confirmed by in situ optical observation of the Ga adlayers on the surface of SLs. For this purpose, we used the effect of decreasing the readouts of the IR-pyrometer by several degrees (~5–7 °C) when a Ga-bilayer appears on the surface, as it has been described by us in [17]. Figure 1a shows the temporal evolution of the IR-pyrometer readouts during the growth of 42 × 6 × {(GaN)_8_/(AlN)_8_} SL with periodic (*K* = 42; *L* = 6) 60-s anneals at a constant power of the substrate heater and closed shutters of all effusion cells and the N_2_***-source. However, at the beginning of each 6 × SL-period growth at time zero in Figure 1a, the readout of IR-pyrometer increases due to radiation heating of the substrate from the open Ga-cell, then it drops sharply to a level below the initial set value of 690 °C. Most likely, this can be associated with the appearance of a Ga bilayer on the GaN surface. In addition, after GaN/AlN switching, a decrease in the temperature can be associated with weaker radiant heating from the Al-cell [17]. A lower level of the readouts persists throughout the growth of both the GaN and AlN layers over six SL periods, and the observed small oscillations (~2 °C) can apparently be explained by different fluxes of thermal radiation from Ga and Al cells, which lead to a small change in the substrate surface temperature. It should be emphasized that during annealing after the growth of six SL periods, the pyrometer readout further falls to a minimum level of about 6 °C below the initial set value, which remains constant and then gradually recovers. Figure 1b shows a similar behavior of the pyrometer readouts during the growth of 44 × 13 × {(GaN)_4_/(AlN)_4_} SL with 44 anneals for 60 s each after every thirteen SL periods. Almost the same duration of the recovery time for the readout of the substrate temperature after the growth of the SL with approximately the same thickness (~26 nm), at least, does not contradict the qualitative model developed above. Thus, the behavior of the IR-pyrometer readouts most likely indicates the periodic formation of small metallic Ga-nanoclusters during the growth of several SL periods, which ensure the continuous occurrence of a Ga-bilayer on the SL surface. The absence of residual traces from metallic GaN droplets on the SL surface was confirmed by their observation with scanning electron and optical microscopes, as shown in Appendix A.

### 3.2. Group-Theoretical Analysis

Short-period wurtzite (GaN)*_m_*/(AlN)*_n_* superlattices (SL’s), though keeping the properties of parent GaN and AlN crystals, are, in fact, new crystals with space groups and atomic arrangements over the Wyckoff positions, which depend on the numbers *m* and *n* of GaN and AlN monolayers per SL primitive unit cell. As was established in our earlier paper [30], short-period (GaN)*_m_*/(AlN)*_n_* SLs grown along the hexagonal symmetry axis constitute two distinct crystal families specified by space groups *P3m1* (# 156) and *P6_3_mc* (# 186) depending on whether *m* + *n* is even or odd, respectively.

In this paper, we have determined the symmetry of phonons in bulk crystals and in the SL family with even *m* + *n* and derived the first-order Raman selection rules. We have also established the genesis of SL phonon modes from those of the bulk.

The phonon symmetry in bulk crystals and in the *P3m1* (# 156) SL family is presented in Table 1 and Table 2, respectively. The phonon symmetries at the symmetry points of the Brillouin zone (BZ) are obtained by the method of site-symmetry induced representations using the SITESYM program [31] of the Bilbao Crystallographic Server (BCS) [32,33,34]. A detailed description of this method can be found in [35].

The structure of Table 1 and Table 2 is the following. Column 1 contains the atomic arrangements over the Wyckoff positions **q**, given in column 2, together with their coordinates in units of basic translations (**a**_1_, **a**_2_, **a**_3_) of the crystallographic unit cell and site symmetry groups G**_q_**. Column 3 gives irreducible representations (irreps) β of G**_q_** according to which the local atomic displacements *x*,*y*,*z* transform. The rest of the columns lists the labels of the little-group irreps induced by the irreps β of G**_q_**. These induced representations determine the symmetries of normal vibrations at the high-symmetry points of the BZ of *P3m1*. The **k**-points are specified by their labels, primitive coordinates, and little co-groups G**_k_** (for more details see Brillouin-zone database on the Bilbao Crystallographic Server [32,33,34]). The notation for space-group irreps follows [36], while that of point-group irreps follows [37]. The description of the space groups and their Wyckoff positions are chosen in correspondence with [38].

From Table 1 and Table 2, one can write down directly the sets of normal phonon modes at any symmetry point of the BZ summarizing the contributions of all the atoms in the primitive unit cell. Those at the symmetry lines can be easily obtained from the compatibility relations.

Each line in Table 1 and Table 2 corresponds to a band representation. All band representations can be also obtained using the program BAND REPRESENTATIONS of the BCS [39,40].

For example, for GaN (AlN) bulk crystals where all atoms occupy 2b positions, there are two phonon bands (see Table 1).

The local *z*-displacements transforming according to a_1_(*z*) induce the following band along the Г–A direction
Г_1_⊕Г_4_–A_1_⊕A_4_
whereas local *x*,*y*-displacements transforming according to e(*x*,*y*) induce the band
Г_5_⊕Г_6_–A_5_⊕A_6_

Note that the A_1_⊕A_4_ and irreps form pairs of complex-conjugated irreps, which correspond to degenerated states with the same energy at the BZ boundary.

For the first-order Raman scattering, the sets of normal modes at the Г-point are of importance. Summarizing the contributions of all the atoms in the primitive cell, we obtain the mechanical representation at the Г-point
Г = Г_o__pt_ + Г_ac_ = 2[Г_1_(*A*_1_) + Г_4_(*B*_1_)+ Г_5_(*E*_2_) + Г_6_ (*E*_1_)],(1)
where Г_ac_ = Г_1_(*A*_1_) + Г_6_ (*E*_1_). Notations of irreps in parentheses follow [37].

The simplest (GaN)_1_/(AlN)_1_ SL is obtained from the bulk GaN crystal by substituting one Ga atom in the primitive unit cell of the wurtzite lattice by the Al one. As a result, two metal atoms in the unit cell, which were connected by symmetry operation, become non-equivalent, and the *P6_3_mc* symmetry of the bulk crystal becomes *P3m1*, whereas the primitive unit cell does not change. In this SL, there are four atoms per primitive cell (Ga, Al, and 2 N).

From Table 2, we write the mechanical representation at the Г point
Г = Г_o__pt_ + Г_ac_ = 4Г_1_(*A*_1_) + 4Г_3_(*E*),(2)
where Г_ac_ = Г_1_(*A*_1_) + Г_3_(*E*).

The Raman selection rules can be obtained using the POLARIZATION SELECTION RULES program [32,33,34] at the BCS. The results are shown in Table 3 and Table 4.

Comparing sets of normal modes for the bulk crystal and the (GaN)_1_/(AlN)_1_ SL, we see that the number of Raman active modes being four in bulk GaN (without taking into account LO-TO splitting) becomes six in the (GaN)_1_/(AlN)_1_ SL with *P3m1* symmetry. It should be noted that for this simplest case of SLs, the primitive unit cell of the SL does not increase with respect to the bulk. Therefore, only point symmetry of the SL decreases, which results in the lifting of the degeneracy of complex-conjugated states at the BZ boundary. The symmetry correspondence between the phonon states of the bulk GaN (AlN) crystal (left side of relations) and that of the (GaN)_1_/(AlN)_1_ SL (right side of relations) at the Г and A points of the BZ has been obtained using the CORREL program of the BCS [32]
Г_1,4_ → Г_1_; Г_5,__6_ → Г_3_; A_1,4_ → A_1_, A_5,6_ → A_3_.(3)

It is seen that Г_1_(*A_1_*) and Г_4_(*B_1_*) modes of bulk GaN transform into Г_1_(*A_1_*) modes of the (GaN)_1_/(AlN)_1_ SL and become non-distinguishable from the point of view of selection rules. In particular, two silent Г_4_(*B_1_*) modes become Raman-active. In turn, Г_5_(*E_2_*) and Г_6_(*E_1_*) modes transform into Г_3_(*E*) ones. Thus, from Equation (3), we see that the phonon modes of the bulk constitute pairs with components transforming into phonon modes of the same symmetry in (GaN)_1_/(AlN)_1_ SL.

For other SLs with the *P3m1* (# 156) symmetry, the corresponding primitive unit cells enlarge, which results in the folding of phonon states at the A-point of the BZ. Thus, the A-point of the BZ turns to be very crucial for the SL phonon genesis.

Using the CORREL program of the BCS [32] with a corresponding unit cell transformation matrix one obtains that the A-point states of the bulk crystals (A_1_⊕A_4_ and A_5_⊕A_6_) transform into the SL states at the Г point as follows: two Г_1_ (*A*_1_) (A_1_ → Г_1_, A_4_ → Г_1_) states with close frequencies and two Г_3_ (*E*) (A_5_ → Г_3_, A_6_ → Г_3_ ) states also with close frequencies.

The SL modes originated from bulk acoustic branches Г_1_–A_1_ (LA), Г_6_–A_6_ (TA) are located in the low and mid-frequency range, whereas those originated from the bulk optical modes lie in the high-frequency range.

The results of the group-theoretical analysis will be used below when discussing the results of the *ab initio* calculations and experiments.

### 3.3. Raman Measurements

#### 3.3.1. Low Energy Phonon Modes

The low energy phonon states in GaN/AlN SLs are closely related to acoustic modes of the bulk materials. The existence of interfaces does not cancel the propagating character of such modes, but changes the sound velocity and gives rise to stationary modes, which are related to the folded acoustic waves. The group-theory analysis of the phonon symmetry in wurtzite-based (GaN)*_m_*/(AlN)*_n_* SLs allows concluding that the folded LA phonons have *A*_1_ and *E* symmetry and can be observed in the $z(xx)\bar{z}$ scattering geometry, and the folded TA phonons have *E* symmetry and can be observed in the $x(yz)\bar{x}$ scattering geometry.

Indeed, in the low-frequency spectral range (0–300 cm^−1^), the SL normal modes are originated from the GaN and AlN bulk acoustic Г_1_–A_1_ (LA), Г_6_–A_6_ (TA) and optical Г_4_–A_4_ (silent), Г_5_–A_5_ phonon branches. As shown above (see Section 3.2), the bulk states A_1_⊕A_4_ and A_5_⊕A_6_ at the BZ boundary correspond to two degenerate phonon states each one combined from one acoustic and one optical phonon branch. In the SLs with even *m* + *n*, this degeneracy is lifted, and the degenerate states when folded transform into two different Г_1_ (*A_1_*) states with close frequencies and two Г_3_ (*E*) states also with close frequencies. Note that in the SL space group, the irreps for all points along the whole Δ-line (Г–Δ–A), including Г and A points, are of the same type, since the group is polar with the polar axis along *z*.

The calculated and experimental low-frequency parts of the $z(xx)\bar{z}$ Raman spectra for 4/4, 6/6, and 8/8 SLs are compared in Figure 2.

One can see that the experimental and calculated $z(xx)\bar{z}$ Raman spectra agree well. This allows one to establish an unambiguous assignment for the observed spectral features. Both spectra contain the doublet-like spectral feature, which exhibits a high-frequency shift while decreasing SL’s period. Additionally, there is another middle-intensity peak that does not move with varying the SL period. An analysis of the calculated eigenvectors allowed us to assign the doublet feature to the folded LA phonons and the middle-intensity Raman peak to the *E*_2_-mode localized within the GaN layers. Eigenvectors of the LA-modes giving rise to the doublet-like Raman feature are displayed in Figure 3.

The most intense peak is related to the LA1 mode. Figure 3a evidences that in this mode, the largest displacement amplitudes correspond to atoms located at the interfaces, and the nodal planes are located in the middle parts of the layers. Such a vibration corresponds to the alternating stress–strain oscillations of the layers: within the SL half-period, the GaN layers expand and AlN layers compress, and vice versa. Anytime, the neighboring layers undergo the *U*_3_ homogeneous deformations of opposite signs. The frequency of such oscillation changes significantly with a change of the SL period because of the difference in the *C*_33_ elastic stiffnesses of the GaN and AlN materials. Relatively high Raman activity of the LA1 mode can be explained by the difference in the *p*_13_ opto-acoustic constants of the GaN and AlN materials. The total value of the derivative $\partial a_{xx}/\partial Q$ is proportional to the difference $p_{13}(\mathrm{GaN})-p_{13}(\mathrm{AlN})$. According to [41], the values of elasto-optical constants *p*_13_ for GaN and AlN crystals differ by about 10%. This difference is sufficient for the LA1 mode to have significant activity in the Raman spectrum in the $z(xx)\bar{z}$ scattering geometry.

Close to the LA1 mode line is the line of the LA2 mode. According to the atomic displacement patterns shown in Figure 3b, this is also the LA-mode with a wavelength equal to the SL period. However, the displacement pattern of this mode is different: the largest amplitudes correspond to the atoms located in the middle of the layers, while the atoms located at the interfaces do not vibrate. For this mode, one half of each layer is compressed, and the other half is stretched. The changes of polarizability caused by the elasto-optical effects in the two halves of every layer have different signs for the LA2 mode. Therefore, they almost cancel each other. As a result, this mode has low Raman activity.

The group-theory analysis of the phonon symmetry of the wurtzite-based (GaN)*_m_*/(AlN)*_n_* SLs with even *m* + *n* suggests that the *E* symmetry modes are active in the Raman spectra in the $x(yz)\bar{x}$ scattering geometry (see Section 3.2). Figure 4 shows the low-frequency parts of the calculated and experimental Raman spectra corresponding to the $x(yz)\bar{x}$ scattering geometry. As was mentioned above, lines corresponding to the TA phonons should be observed in this case.

It can be seen that in calculations and experiments, the line positions noticeably change depending on the SL period. The proximity of the positions of these peaks in the calculations and experiment and the similarity of their dependence on the structure allows us to suggest that the lines in the calculated and observed spectra correspond to the same phonons. To understand the nature of these modes, we turn to the analysis of the calculated eigenvectors. According to the calculation, the lowest frequency line in the $x(yz)\bar{x}$ spectrum is related to the TA modes with a wavelength equal to the SL period. The second spectral line at around 30 cm^−1^ higher (labeled as 2TA) corresponds to the TA acoustic wave with a twice shorter wavelength.

As it was noted in [42], the frequency position of the TA and 2TA lines carries information that allows one to determine the period of the SL. In the elastic continuum approximation, the frequencies of the acoustic TA and 2TA phonons are determined by the relations:(4)$$\nu (\mathrm{TA})=V_{\mathrm{SL}}/d,\;\nu (2\mathrm{TA})=2V_{\mathrm{SL}}/d$$
where *V*_SL_ is the speed of sound in the SL and *d* is the period of the SL.

Our computational and experimental results confirm the validity of Equation (4) with *V*_SL_ = 4.66 km/s. This value agrees well with the equation $\frac{d}{V_{\mathrm{SL}}}=\frac{d_{1}}{V_{1}}+\frac{d_{2}}{V_{2}}$, which relates the sound velocity in the SL with the velocities in GaN and AlN crystals [43,44].

#### 3.3.2. E(TO) Phonons

Already in early works devoted to the study of phonons in SLs [16,42], it was established that for polar phonons with a wave vector perpendicular to the interface plane, the dipole–dipole interaction between the vibrations of atoms of neighboring layers is completely screened by surface charges at the interfaces. Therefore, in SLs made of materials with strongly differing phonon frequencies, such modes have the character of vibrations localized in separate layers. Nonpolar modes are also localized if their frequencies in one material do not overlap with the frequencies of similar modes in the materials of adjacent layers. In the case of binary GaN/AlN SLs, the first type includes modes that correspond to polar *E*_1_(TO) vibrations in bulk GaN and AlN wurtzite crystals, and the second one—to nonpolar vibrations of the *E*_2_ type. The *E*_2_*–E*_1_ frequency intervals in GaN and AlN do not overlap. Therefore, the corresponding SL modes have the character of stacked standing waves localized in separate layers.

Although in nitride SLs the GaN and AlN layers retain the packing symmetry of the wurtzite structure, due to the disappearance of the sixth order screw axis, the modes of the *E*_1_ and *E*_2_ irreps transform into the modes with the same symmetry described by the *E* irrep (see Section 3.2). In the Raman spectra of the $z\left(xy\right)\bar{z}$ geometry, modes of this type give groups of lines in the *E*_2_*–E*_1_ frequency ranges of GaN and AlN. The number of lines in each such group should correspond to the number of monolayers in the GaN and AlN layers. Note that in both materials, the frequency intervals *E*_2_–*E*_1_ are relatively narrow (~10 cm^−1^). Therefore, in an experiment, we are unlikely to be able to distinguish between individual lines and determine their number. Rather, these lines will merge into an asymmetric band ~20–30 cm^−1^ wide. The asymmetry of this band is a consequence of the different Raman intensities of the *E*_1_ and *E*_2_ modes. In both materials, the *E*_2_ modes show noticeably higher Raman activity. Taking into account that the ω(*E*_1_) < ω(*E*_2_) ratio is observed in GaN and in AlN the opposite ω(*E*_2_) < ω(*E*_1_) holds, we can expect that the maximum in the *E*(GaN) band contour will be closer to the high-frequency edge, and in the *E*(AlN) band contour, on the contrary, the maximum will be closer to the low-frequency edge.

It is these bands that we observe in the spectra of the studied SLs. Figure 5 shows the $z\left(xy\right)\bar{z}$ geometry Raman spectra of (GaN)*_m_*/(AlN)*_n_* (*m*, *n* = 4,6,8) SLs in the frequency range of *E*(TO) oscillations.

Both in the experiment and the calculation, two asymmetric bands are clearly distinguishable in each spectrum, which correspond to the *E*(TO) vibrations localized in the GaN and AlN layers. This assignment can be confirmed by analyzing the corresponding eigenvectors. Consider, for example, the spectrum of the (GaN)_6_/(AlN)_6_ SL. In Figure 6, the experimental spectrum of the $\;z\left(xy\right)\bar{z}$ geometry is compared with the calculated one.

It can be seen that six modes contribute to the profile of every band: modes 1–6 to the *E*(TO)-GaN band and modes 7–12 to the *E*(TO)-AlN band. The eigenvectors of the extreme ones are shown in Figure 7.

The presented displacement patterns show that modes 1 and 6 (like the rest 2–5) correspond to *E*(TO) vibrations localized in the GaN layer, and modes 7 and 12 (like the other 8–11) correspond to *E*(TO) vibrations localized in the AlN layer. Moreover, modes 1 and 12 are similar to the *E*_1_(TO) modes in bulk crystals: all cations move in phase in the direction opposite to the movement of the anions. In turn, modes 6 and 7 are similar to the *E*_2_ modes in bulk crystals: neighboring cations (like anions) move in opposite directions. This analysis confirms our assignment of the *E*(TO) line edges to the *E*_1_ and *E*_2_ modes.

The appearance of doublets in the Raman spectra in the frequency range of *E*(TO) phonons can be also explained based on the results of the group-theoretical analysis.

The SL *E*(TO) modes originate from the pair of bulk phonon branches:
Г_5_–A_5_ (*E*_2_(TO)-branch) and Г_6_–A_6_ (*E*_1_-branch),
which stick together at the A-point of the BZ, i.e., they form a pair of complex-conjugated irreps A_5_⊕A_6_, which correspond to a degenerate phonon state (see Section 3.2). In SLs, this degeneracy is lifted due to the lowering of space symmetry from *P*6*_3_mc* (# 186) to *P3m1* (# 156). As a result, this degenerate state splits at the A-point and then transforms into the Г-states of the SL due to folding: A_5_ → Г_3_, and A_6_ → Г_3_. As a result, the small splitting of the degenerate bulk states at the BZ boundary leads to the formation of *E*(TO) doublets at the Г-point in SLs. It should be noted that the frequencies of the *E*_2_ and *E*_1_ modes in GaN and AlN are very sensitive to lattice strains. This dependence has become the subject of numerous studies [45,46,47,48,49], the result of which has been the determination of the so-called deformation potentials that relate frequency shifts of the Raman lines with the homogeneous strain of the crystals. 

The observed shifts of the lines in Figure 6 relative to their position in bulk unstressed crystals are definitely associated with the strains of the layers that arise during the growth of the heterostructure. In particular, the line shifts corresponding to the localized *E*(TO) vibrations in the AlN and GaN layers have different signs that indicate different signs of strains in these layers. In Table 5, the in-plane strain values in the SL layers are summarized, which were obtained using the data on the phonon deformation potentials for GaN [46] and AlN [49].

The obtained estimations of the elastic strains are based on the deformation potentials derived from investigations of rather thick epitaxial layers or bulk crystals. Whether they apply to the short-period SLs, the answer to this question will be given after additional experimental studies of the structural characteristics of the SLs samples. This issue is under study.

#### 3.3.3. A_1_(TO) Phonons

As was established earlier [16], the polar phonons with a wave vector directed within the interface plane have a delocalized character. Due to the dipole–dipole interaction, atoms of all layers are involved in such vibrations. Consequently, the frequency value depends on the ratio of the layer thicknesses, varying in the range of values corresponding to bulk materials. This circumstance makes such modes a convenient tool for monitoring the composition of SLs.

In the spectra of binary nitride SLs, the polar delocalized modes include *A*_1_(TO) and *E*_1_(LO) symmetry modes. The *A*_1_(TO) modes are the most favorable for controlling the composition, since they give intense lines in the Raman spectra in the $x\left(zz\right)\bar{x}$ scattering geometry, which are convenient for observation. Examples of such spectra of the samples studied in this work are shown in Figure 8.

It can be seen that the calculation quite accurately reproduces the experimentally observed spectral curves with a slight systematic overestimation of the frequency magnitude by about 5 cm^−1^. The calculation also confirms the delocalized nature of these modes (see Figure 9). Analysis of the calculated eigenvectors of these modes allows us to understand the nature of their high intensity in the spectra. The pattern of atomic displacements is similar to that of the TO modes in pure crystals, and the amplitudes of both cations and anions in all layers are approximately the same.

As was predicted in early papers [16,50], the frequency of *A*_1_(TO) modes in SLs should strongly depend on the ratio of the layer thicknesses. The analysis of the spectra of the samples studied in this work revealed that the dependence of the frequency on the composition parameter x=n/(m+n) turned out to be close to linear (see Figure 10). This result differs from that obtained in our previous works [16] based on the study of SLs grown by MOVPE. The *A*_1_(TO) mode frequencies for those samples (also shown in Figure 10) are in a good agreement with the ω(*x*) dependence established for solid solutions [51] and predicted by the dielectric continuum model [16].

It should be noted that the ω(*x*) dependence, based on the results of non-empirical calculations, also has a rather non-linear character. Moreover, as the total number of layers in the SL period increases, the calculated ω(*x*) dependence progressively approaches the experimental one.

The difference in the dependences established in this case for two types of SLs can be explained by the difference in the quality of the samples grown by different methods. According to structural studies, the samples studied in this work have well-localized interfaces. That is why the ω(*x*) dependence determined for them is in good agreement with the results of calculations, in which the interfaces were assumed to be abrupt.

#### 3.3.4. A_1_(LO) Phonons

As well as the *E*(TO) modes considered above, the polar modes with symmetry *A*_1_(LO) have the character of vibrations localized in separate layers of GaN/AlN SLs. The bands in Raman spectra that correspond to these vibrational modes should be observed in the spectra obtained in the $z\left(xx\right)\bar{z\;}$ scattering geometry in the ranges 690–733 (GaN-like) and 719–875 (AlN-like) cm^−1^, since the frequency ranges of these modes for bulk GaN and AlN crystals slightly overlap. The calculated and experimental high-frequency Raman spectra for the $z\left(xx\right)\bar{z}$ scattering geometry, obtained for some of the studied SLs, are shown in Figure 11. It can be seen that there is a good agreement between the calculated and experimental spectra. This allows us to make an unambiguous assignment of the observed spectral features. For localized phonons, the frequency values ω and the number of such modes should depend on the individual characteristics of the layers comprising the SL. The latter circumstance makes such modes a convenient tool for monitoring the SL structure. Let us show that the number of *A*_1_(LO) symmetry modes depends on the thickness of the SL layers. For this, we consider in more detail the spectrum in the $z\left(xx\right)\bar{z}$ geometry for the (GaN)_8_/(AlN)_8_ SL shown in Figure 12.

In the range of 740–890 cm^−1^, there are eight modes in the spectrum, half of which give a noticeable contribution in the Raman spectrum. The atomic displacements in the corresponding modes are shown in Figure 13. It can be seen that (a) the atoms in the AlN layers are mainly involved in these vibrations; (b) these vibrations really correspond to the shape of *A*_1_(LO) modes (cations and anions are shifting along the *z*-axis in opposite directions); (c) the amplitude distribution (at least for the first five modes) resembles the harmonics of standing waves with nodes at the interfaces.

The latter circumstance explains the fact that only half of these modes have the Raman scattering intensity noticeably different from zero. These are odd-numbered modes (Figure 13, 1,3,5,7), which correspond to harmonics with a non-integer number of periods in the layer. In the experimental Raman spectrum (shown by the blue line in Figure 11), the four well-pronounced bands, which correspond to the above-mentioned vibrations, can be distinguished, although the line of mode 7 overlaps strongly with the wide band of the *A*_1_(LO) GaN mode. Thus, we conclude that the presence of *N* lines in the $z\left(xx\right)\bar{z}$ spectrum in the 740–890 cm^−1^ range indicates that the AlN layers under study contain 2*N* monolayers. This conclusion is also confirmed by the spectra of other SLs studied shown in Figure 11: there are two bands corresponding to the *A*_1_(LO)-AlN modes in the Raman spectra of 4/4 SL, and three bands in the spectrum of 6/6 SL.

In Figure 12, one can also see the presence of eight modes in the range of 700–740 cm^−1^. These are *A*_1_(LO) modes localized in GaN layers. Due to the narrowness of this frequency interval, it is not possible to distinguish between the corresponding lines (or even determine their number) in the experimental spectrum. In the experiment, all these bands merge into one strong asymmetric peak with a maximum close to *A*_1_(LO)-GaN. An analysis of the eigenvectors of the *A*_1_(LO) modes localized in the GaN layers showed that they all noticeably mixed with the *B*_1_ mode of the AlN layer and therefore, strictly speaking, are not localized.

A question arises: what does the Raman spectrum of the *A*_1_(LO)-mode look like for an odd number? Let us consider this issue on the example of (GaN)*_m_*/(AlN)_12-*m*_ SL. Figure 14a–c shows the calculated and experimental spectra in the $z\left(xx\right)\bar{z}$ geometry in the range of *A*_1_(LO) modes of three SLs: 7/5, 6/6, and 5/7. The bars show individual modes and represent the Raman intensity. On the whole, we can state a good agreement between the calculation and experiment, both in the number of peaks and in their position and relative intensity. It can also be seen that in the 740–900 cm^−1^ range, the 7/5, 6/6, and 5/7 SL spectra contain contributions from five, six, and seven modes, respectively. This is consistent with the number of standing wave harmonics localized in the AlN layers. However, as noted above, only three out of six modes make a noticeable contribution to the spectrum of the 6/6 equal-period SL. In the 7/5 SL, there are three of five such modes, and in the 5/7 SL, there are four of seven such modes. Moreover, the last odd harmonics (the fifth in 7/5 SL and the seventh in the 5/7 SL) have a high Raman intensity comparable to the intensity of the first harmonic. This effect can be explained by analyzing the eigenvectors of these modes. Both of them noticeably mix with atomic vibrations in the GaN layers, and these admixed vibrations are symmetric with respect to the middle of the GaN layer.

Based on the results of the group-theoretical analysis, we can establish the hierarchy of *A*_1_(LO) mode intensities in Raman spectra. Similarly, like E-branches, the *A*_1_(LO) modes in SL’s originate from the pair of bulk phonon branches:Г_1_–A_1_ (*A*_1_(LO)-branch) and Г_4_–A_4_ (*B*_1_—silent branch).

These branches also stick together at the A-point of the BZ, i.e., they form a pair of complex-conjugated irreps A_1_⊕A_4_, which correspond to a degenerate phonon state (see Section 3.2). In SLs, due to the lowering of space symmetry, this degenerate state at the A-point splits and then, due to folding, transforms into the Г-states of the SL: A_1_ → Г_1_ [*A*_1_(LO)], A_4_ → Г_1_ [*A*_1_(LO)]. In turn, the Г_4_ [*B*_1_] state also transforms into the Г_1_ [*A*_1_(LO)] mode. Thus, among the Г_1_ [*A*_1_(LO)] modes in the SL, there are strong modes originated from the bulk Г_1_–A_1_ (*A*_1_(LO)-branch) and relatively weak modes originated from the bulk Г_4_–A_4_ (*B*_1_ -silent branch).

Thus, we can extend the scheme for interpreting the $z\left(xx\right)\bar{z}$ geometry spectra of short-period (GaN)*_m_*/(AlN)*_n_* SLs in the *A*_1_(LO) frequency range to the case of odd *n*. The following rules can be suggested:If the line with the lowest frequency in the 740–900 cm^−1^ interval has relatively low intensity (less than the intensity of the first line) and does not overlap with the *A*_1_(LO)-GaN line, then in the studied SL, the number *n* is even and is twice the number of lines observed in this spectrum.If this last line has a high intensity (comparable to that of the first line) and is close in frequency to the *A*_1_(LO)-GaN line (which manifests itself as a doublet splitting of this line), then in the SL under study, the number *n* is odd and equal to 2*k* + 1, where *k* is the number of lines observed in this spectrum.

The results presented in this Section can be summarized as follows. The established close-to-linear dependence of the *A*_1_(TO) frequency on the concentration parameter x=n/(m+n) together with the data on the SL period derived from the frequency of the folded acoustic LA1 phonon can be used to quantify the thicknesses of the GaN and AlN layers. Additional verification of the estimated thickness of the AlN layer is a determination of the number of monolayers, which is based on an analysis of *A*_1_(LO) Raman peaks in the LO(AlN) spectral region. The latter method was first proposed in this work.

The experimental Raman spectra for the 4/4, 6/6, and 8/8 SLs are shown in Appendix A in order to demonstrate the changes in the intensities and positions of the acoustic and optical modes in dependence of the SL period.

## 4. Summary and Conclusions

The dynamical properties of short-period AlN*_m_*/GaN*_n_* SLs (*m* + *n* ≤ 16) SLs grown by the submonolayer digital PA MBE method were studied. A comprehensive group-theoretical analysis made it possible to establish the genesis of the SL phonon modes from the modes of bulk AlN and GaN crystals. A symmetry relationship was established between the vibrations of the bulk GaN (AlN) crystal and the SL’s at the high-symmetry points of the Brillouin zone. Based on the genesis of the phonon modes, the conclusions on the frequencies and intensities of the corresponding bands in the Raman spectra of the SL have been made. *Ab initio* calculations in the framework of the density functional theory, aimed to study the phonon states, were performed for SLs with both equal and unequal layer thicknesses. As a result, the phonon mode frequencies were calculated, and the atomic displacement patterns were established. The number and symmetry of the vibrational modes in the calculated spectrum are in good agreement with the results of the group-theoretical analysis. The Raman tensor was calculated within density functional perturbation theory for the series of SLs, and simulated Raman spectra were compared with experimental ones. The results of the *ab initio* calculations are found to be in good agreement with the experimental Raman spectra.

The doublet structure in the low-frequency range of the Raman spectra is explained by the presence of two types of folded LA phonons, which differ in the position of the nodal planes. For one type of modes, the nodal planes coincide with the interfaces, whereas for other modes, they are located approximately in the middle of the SL layers. The dependences of the positions and intensities of the folded LA and TA bands in the Raman spectra on the period of the SL are determined. It was found that the frequencies of the folded acoustic bands depend not only on the total period of the SL, but also on the ratio of layer thicknesses, which opens up new possibilities for the diagnostics of the SL structure. It has been established that the *E*(TO) modes are localized in the constituent SL layers. The doublet structure of the *E*(TO) lines localized in the GaN and AlN SL layers genetically related to the *E*_2_(high)- and *E*_1_- branches of the bulk crystal was first revealed and explained. The results can be used to obtain information about the strain in each layer forming the SL. In turn, it was confirmed that the *A*_1_(TO) mode has a delocalized nature. It was established that the ω*_A_*_1(TO)_ frequency value can be used to estimate such an important SL composition parameter as x=n/(m+n). It was found that, in the high-frequency region of the spectrum, the modes originating from the polar *A*_1_(LO)-branch folded from the A-point at the BZ boundary have a higher intensity than the accompanying satellite lines originating from the silent *B*_1_-branch also folded from the A-point. The calculations revealed a strong dependence on the layer thickness of the number and intensities of the modes localized in AlN layers forming the SL. An approach was proposed for estimating the number of AlN monolayers forming SL, both for equal and unequal thicknesses of constituent GaN and AlN layers of the SLs.

As a result of comprehensive studies, the correlations between the parameters of acoustic phonons and localized and delocalized optical phonons and the structure of SLs were obtained. This opens up new possibilities for analyzing the structural characteristics of short-period GaN/AlN SLs using Raman spectroscopy. The results obtained can be used to optimize the parameters of the growth process in order to form structurally perfect short-period GaN/AlN SLs with different ratios of the thicknesses of the layers forming the SL, as well as digital AlGaN alloys grown on the basis of related nanostructures.

## Figures and Tables

**Figure 1 nanomaterials-11-00286-f001:**
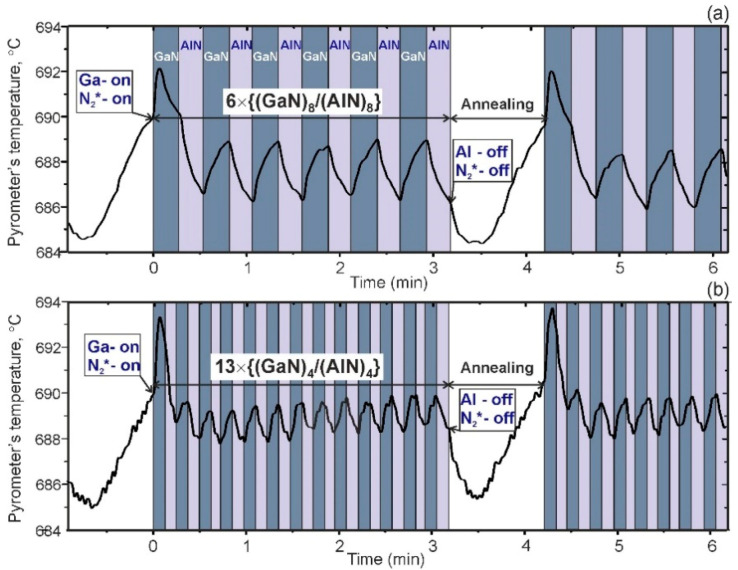
Temporal evolutions of the IR-pyrometer readouts during the growth of 42 × 6 × {(GaN)_8_/(AlN)_8_} (**a**) and 44 × 13 × {(GaN)_4_/(AlN)_4_} (**b**) SLs with the same growth parameters and the same duration of short-term annealing after growth of every six and thirteen SL periods, respectively.

**Figure 2 nanomaterials-11-00286-f002:**
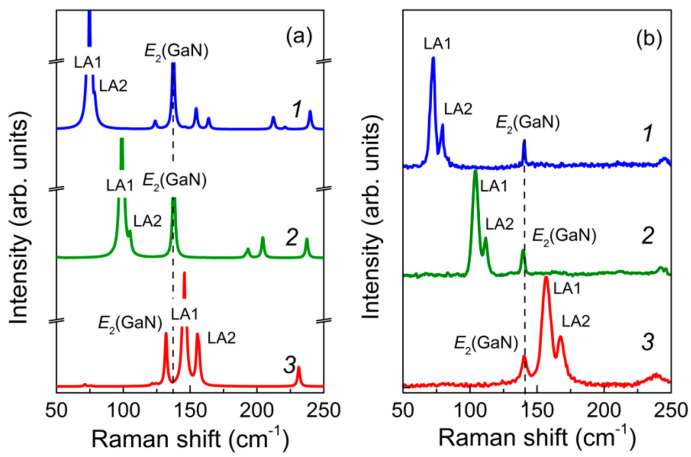
The low-frequency parts of the calculated (**a**) and experimental (**b**) Raman spectra in the $z(xx)\bar{z}$ scattering geometry for the (GaN)*_m_*/(AlN)*_n_* SLs with *m*/*n* = 8/8 (*1*), 6/6 (*2*), and 4/4 (*3*).

**Figure 3 nanomaterials-11-00286-f003:**
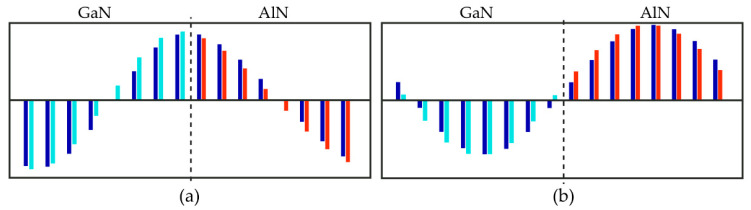
The atomic displacement patterns of longitudinal LA1 (**a**) and LA2 (**b**) phonon modes. Cyan, red, and blue bars denote Ga, Al, and N atomic displacements along the *c*-axis, respectively.

**Figure 4 nanomaterials-11-00286-f004:**
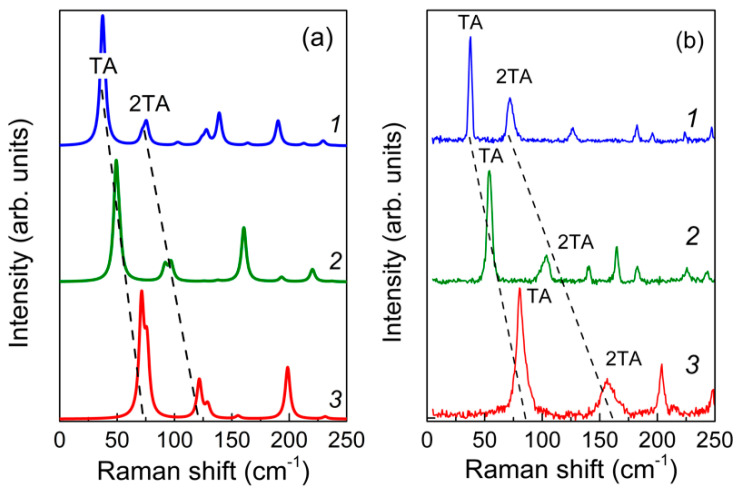
The low-frequency parts of the calculated (**a**) and experimental (**b**) Raman spectra in the $x(yz)\bar{x}$ scattering geometry for the (GaN)*_m_*/(AlN)*_n_* SLs with *m*/*n* = 8/8 (*1*), 6/6 (*2*), and 4/4 (*3*).

**Figure 5 nanomaterials-11-00286-f005:**
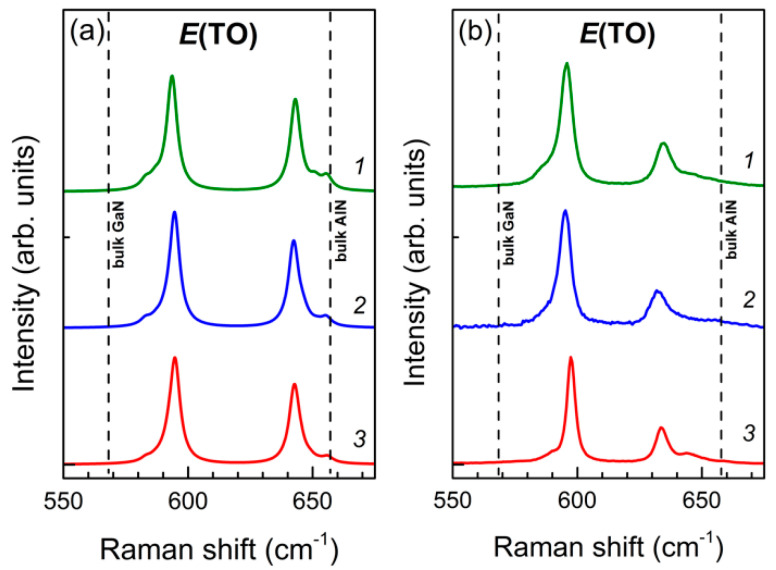
Calculated (**a**) and experimental (**b**) Raman spectra of (GaN)*_m_*/(AlN)*_n_* SLs in the $z\left(xy\right)\bar{z}$ scattering geometry in the region of *E*(TO) modes, *m*/*n*: *1*—4/4, *2*—6/6, *3*—8/8.

**Figure 6 nanomaterials-11-00286-f006:**
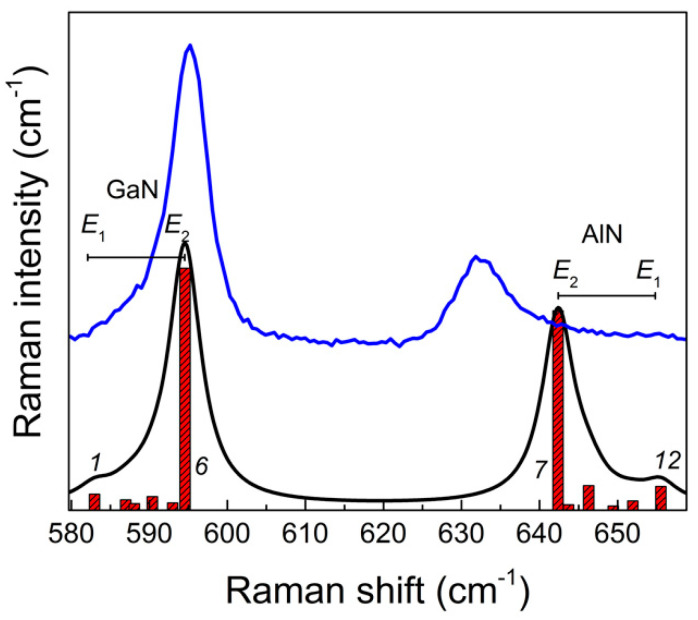
Raman frequencies and intensities of the *E*(TO) symmetry modes in the calculated spectrum of the $z\left(xx\right)\bar{z}$ geometry for the (GaN)_6_/(AlN)_6_ SL. The solid black line shows the averaged spectrum for the Lorentzian line shape and the line half-width of 3 cm^−1^. The blue line shows the experimental spectrum.

**Figure 7 nanomaterials-11-00286-f007:**
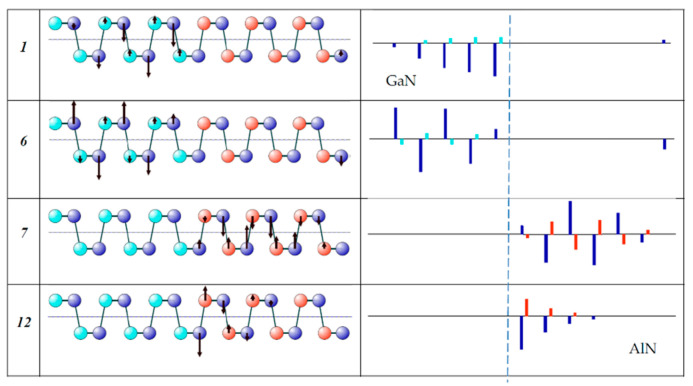
Eigenvectors of the *E*(TO) modes (*x*-displacement of atoms) corresponding to the edges of the *E*(TO)-GaN (1,6) and *E*(TO)-AlN (7, 12) lines. Atomic displacements for the high (a,c) and low (b,d) intensity modes *E*(GaN) (left column) and *E*(AlN) (right column) in (GaN)_6_/(AlN)_6_ SL.

**Figure 8 nanomaterials-11-00286-f008:**
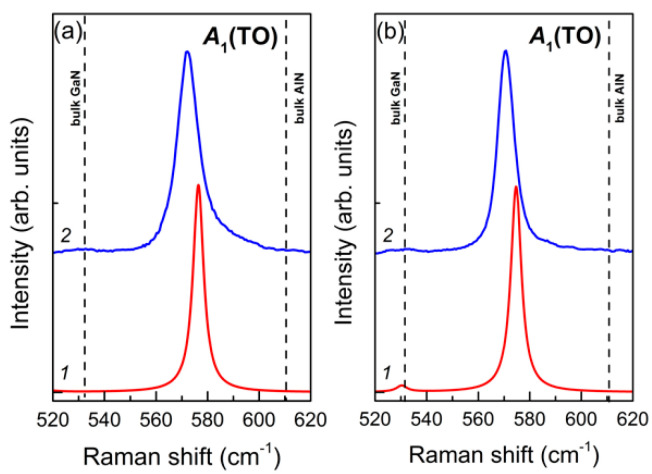
Calculated (*1*) and experimental (*2*) Raman spectra of (GaN)*_m_*/(AlN)*_n_* SLs in the $x\left(zz\right)\bar{x}$ scattering geometry in the region of *A*_1_(TO) modes, *m*/*n*: 4/4 (**a**), 6/6 (**b**).

**Figure 9 nanomaterials-11-00286-f009:**
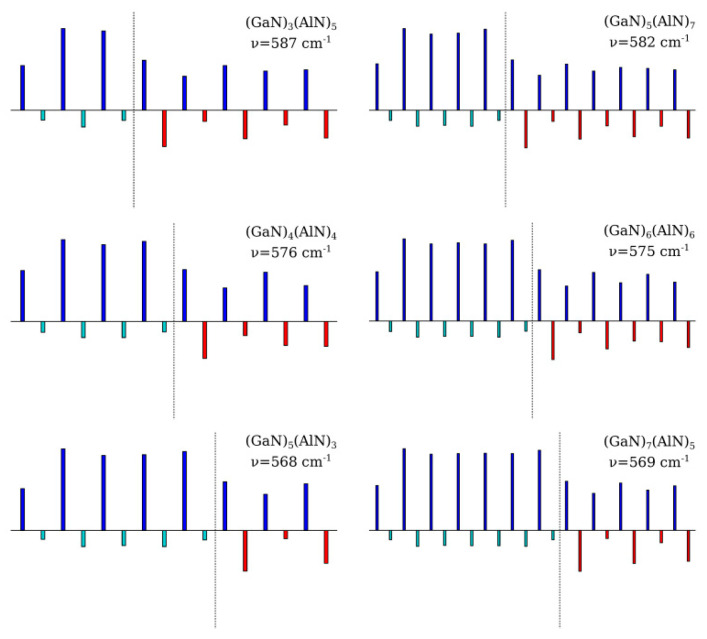
Calculated atomic *z*-displacement amplitudes for the delocalized *A*_1_(TO) modes of the (GaN)*_m_*/(AlN)*_n_* SLs: blue, cyan, and red bars correspond to displacements of N, Ga, and Al atoms, respectively.

**Figure 10 nanomaterials-11-00286-f010:**
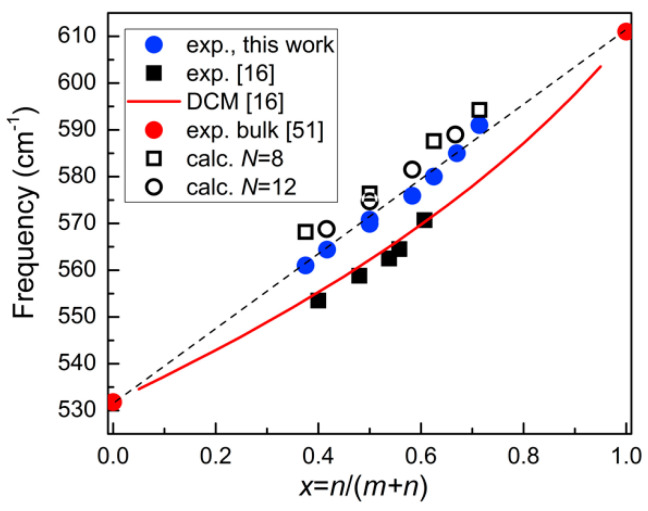
Calculated and experimentally determined frequencies of *A*_1_(TO) modes in (GaN)*_m_*/(AlN)*_n_* SLs in dependence of the composition parameter x=n/(m+n). Computational results are shown by open symbols, experimental results by blue circles (this study), and black squares (data from [16]). Red circles indicate *A*_1_(TO) experimental frequencies in bulk crystals. The ω(*x*) dependence predicted by the dielectric continuum model is shown by the red line.

**Figure 11 nanomaterials-11-00286-f011:**
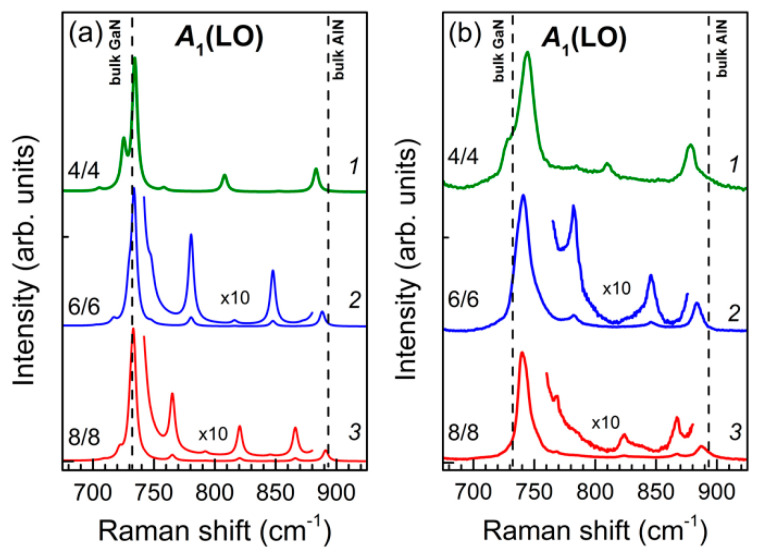
The high-frequency parts of the calculated (**a**) and experimental (**b**) $z(xx)\bar{z}$ Raman spectra for the (GaN)*_m_*/(AlN)*_n_* SLs with *m*/*n* = 4/4 (*1*), 6/6 (*2*) and 8/8 (*3*).

**Figure 12 nanomaterials-11-00286-f012:**
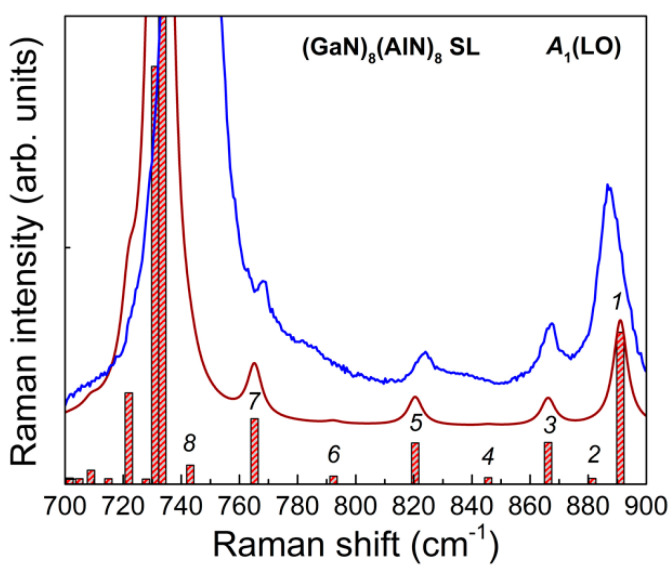
Frequencies and Raman intensities of the *A*_1_(LO) symmetry modes in the calculated spectrum in the $z\left(xx\right)\bar{z}$ geometry for (GaN)_8_/(AlN)_8_ SLs (shown by bars). The solid brown line shows the averaged spectrum for the Lorentzian line shape and the line half-width of 3 cm^−1^. The blue line shows the experimental spectrum.

**Figure 13 nanomaterials-11-00286-f013:**
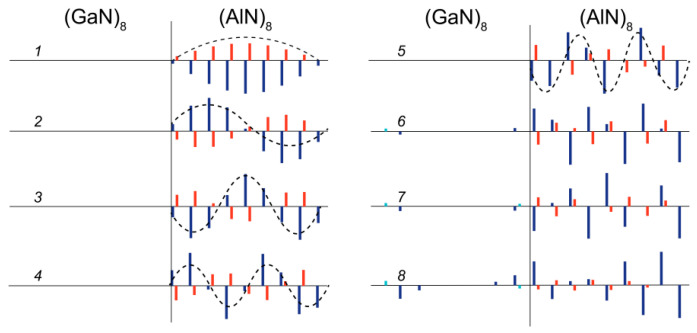
Eigenvectors of the *A*_1_(LO) modes in the (GaN)_8_/(AlN)_8_ SLs.

**Figure 14 nanomaterials-11-00286-f014:**
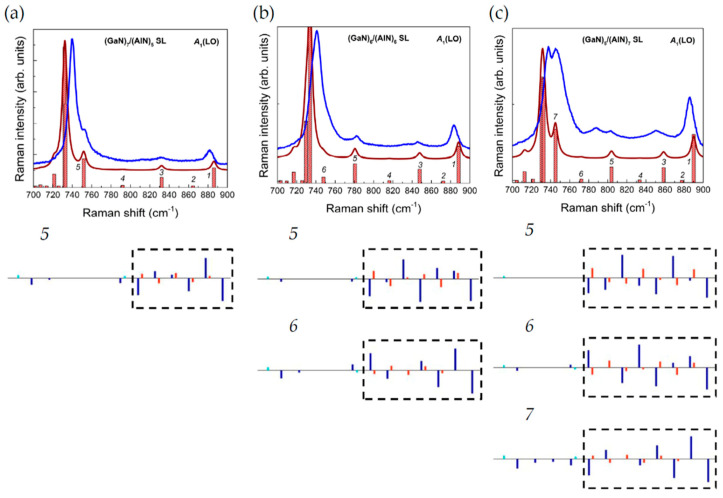
Frequencies and Raman intensities of the *A*_1_(LO) symmetry modes in the calculated spectrum in the $z\left(xx\right)\bar{z}$ geometry for (GaN)_7_/(AlN)_5_ (**a**), (GaN)_6_/(AlN)_6_ (**b**), and (GaN)_5_/(AlN)_7_ (**c**) SLs (shown by bars). The solid brown line shows the averaged spectrum for the Lorentzian line shape and the line half-width of 3 cm^−1^. The blue line shows the experimental spectrum. Below the spectra, the eigenvectors of some modes are given, the numbers of which correspond to the numbering shown on the panels (**a**,**b**,**c**). Dashed lines show the AlN layers in SLs.

**Table 1 nanomaterials-11-00286-t001:** Phonon symmetries in the bulk GaN (AlN) with the space group *P6_3_mc*. Pairs of conjugated irreps: A_1_⊕A_4_, A_5_⊕A_6_, H_1_⊕H_2_, L_1_⊕L_4_, L_2_⊕L_3_.

Atoms	q	β	Г	A	K	H	M	L
			(000)	(00$\frac{1}{2}$)	($\frac{1}{3}$$\frac{1}{3}$0)	($\frac{1}{3}$$\frac{1}{3}$$\frac{1}{2}$)	($\frac{1}{2}$00)	($\frac{1}{2}$0$\frac{1}{2}$)
			*6mm*	*6mm*	*3m.*	*3m.*	*2mm*	*2mm*
2Ga(Al) 2N	2*b* ($\frac{1}{3}$$\frac{2}{3}$*z*) ($\frac{2}{3}$$\frac{1}{3}$*z*+$\frac{1}{2}$) 3*m*.	*a_1_(z)* *e(x,y)*	1,4 5,6	1,4 5,6	3 1,2,3	3 1,2,3	1,4 1,2,3,4	1,4 1,2,3,4

**Table 2 nanomaterials-11-00286-t002:** Phonon symmetries in the (GaN)*_m_*/(AlN)*_n_* SL’s with the space group *P**3m1* (*m* + *n* = 2k).

Atoms		q	β	Г	A	K	H	M	L
*m* = 2s + 1 *n* = 2t + 1	*m* = 2s *n* = 2t			(000)	(00$\frac{1}{2}$)	($\frac{1}{3}$$\frac{1}{3}$0)	($\frac{1}{3}$$\frac{1}{3}$$\frac{1}{2}$)	($\frac{1}{2}$00)	($\frac{1}{2}$0$\frac{1}{2}$)
				*3.m*	*3.m*	*3..*	*3..*	*..m*	*..m*
$\frac{m+1}{2}$Ga $\frac{n-1}{2}$Al $\frac{m+n}{2}$N	$\frac{m}{2}$Ga $\frac{n}{2}$Al $\frac{m+n}{2}$N	1*b* ($\frac{1}{3}$$\frac{2}{3}$*z*) 3*m*.	*a_1_(z)* *e(x,y)*	1 3	1 3	2 1,3	2 1,3	1 1,2	1 1,2
$\frac{m-1}{2}$Ga $\frac{n+1}{2}$Al $\frac{m+n}{2}$N	$\frac{m}{2}$Ga $\frac{n}{2}$Al $\frac{m+n}{2}$N	1*c* ($\frac{2}{3}$ $\frac{1}{3}$*z*) 3*m*.	*a_1_(z)* *e(x,y)*	1 3	1 3	3 1,2	3 1,2	1 1,2	1 1,2

**Table 3 nanomaterials-11-00286-t003:** The POLARIZATION SELECTION RULES [32,33,34] at the BCS for the Raman spectra of the bulk GaN (AlN) crystals. In the table, “×” represents the modes that can be observed in each one of the directions. In the first row, the experimental configuration in Porto’s notation is given.

	$\bar{x}\left(yy\right)x$	$\bar{x}\left(yz\right)x$	$\bar{x}\left(zz\right)x$	$\bar{y}\left(xx\right)y$	$\bar{y}\left(xz\right)y$	$\bar{y}\left(zz\right)y$	$\bar{z}\left(xx\right)z$	$\bar{z}\left(xy\right)z$	$\bar{z}\left(yy\right)z$
*A*_1_(LO)							×		×
*A*_1_(TO)	×		×	×		×			
*E*_1_(TO)		×			×				
*E* _2_	×			×			×	×	×

**Table 4 nanomaterials-11-00286-t004:** The POLARIZATION SELECTION RULES [32,33,34] at the BCS for the Raman spectra of the (GaN)*_m_*/(AlN)*_n_* SL family with even *m* + *n* in the backscattering geometry. In tables, “×” represents the modes that can be observed in each one of the directions. In the first row, the experimental configuration in Porto’s notation is given.

	$\bar{x}\left(yy\right)x$	$\bar{x}\left(yz\right)x$	$\bar{x}\left(zz\right)x$	$\bar{y}\left(xx\right)y$	$\bar{y}\left(xz\right)y$	$\bar{y}\left(zz\right)y$	$\bar{z}\left(xx\right)z$	$\bar{z}\left(xy\right)z$	$\bar{z}\left(yy\right)z$
*A*_1_(LO)							×		×
*A*_1_(TO)	×		×	×		×			
*E*(LO)				×					
*E*(TO)	×	×			×		×	×	×

**Table 5 nanomaterials-11-00286-t005:** Estimated values of in-plane strain *ε_xx_* in the (GaN)*_m_/*(AlN)*_n_* SLs. The negative sign of *ε_xx_* corresponds to the compressive strain.

Samples, *m*/*n*	GaN			AlN		
	*ω*(*E*_2_), cm^−1^	Δ*ω*(*E*_2_), cm^−1^	*ε_xx_*, 10^−2^	*ω*(*E*_2_), cm^−1^	Δ*ω*(*E*_2_), cm^−1^	*ε_xx_*, 10^−2^
Bulk	567.8	0	0	657.4	0	0
3/5	599.5	31.7	−2.61	644.8	−12.6	+0.66
4/4	595.7	27.9	−2.30	634.7	−22.7	+1.18
5/3	580.8	13.0	−1.07	616.6	−40.8	+2.13
5/7	600.0	32.2	−2.66	638.2	−19.2	+1.00
6/6	595.2	27.4	−2.26	632.4	−25.0	+1.30
7/5	593.0	25.2	−2.08	628.4	−29.0	+1.51
8/8	596.9	29.1	−2.40	633.6	−23.8	+1.24

## Data Availability

The data presented in this study are available on request from the corresponding author.

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
