# Peer review of "Phonons in Short-Period GaN/AlN Superlattices: Group-Theoretical Analysis, Ab initio Calculations, and Raman Spectra"

_nanomaterials, 2021, doi:10.3390/nano11020286_

Round 1

Reviewer 1 Report

This manuscript reports on a detailed characterization of GaN/AlN heterostructure superlattices (SL) based on measured Raman spectra and calculated phonon frequencies. A very good agreement is found between measured and simulated Raman spectra, which allowed a precise assignment of the measured pick to the phonon symmetry, its nature (localized/delocalized), and its dependence on the structural parameters, such that the number of layers and the superlattice period.

The analysis reported in this paper goes from the synthesis of the SL to the phonon spectrum study. Its extension is impressive. The paper is well organized and very well written. It could have a significant impact to the community, as it determines to which extent measured by Raman spectroscopy could reveal the heterostructure geometry and composition of the SL.

I recommend therefore publication, provided the authors take care of these (minor) points:

1) The simulated spectra are plotted with some Lorentzian broadening. What is the value of the broadening parameter? This detail should be reported in the text.

2)In Figure 3, LA patterns are plotted. One should specify that these phonons are longitudinal indeed. By looking at the Figure, one can imagine that they are transverse.

3) In page 12, it is written that "This dependence has become the subject of numerous studies." The authors should cite these studies, at least the most important ones.

4) It would be desirable to make a Figure showing all modes together, to get a clearer idea on the relative intensity of the different modes, and their relative position.

Typo: Bottom of page 10. The parenthesis in "geometry)" should be removed.

Author Response

Dear Reviewer,

We are grateful to you for the careful reading of our manuscript and valuable comments you made. We have addressed all your points and made corresponding corrections in the text of the manuscript:

1) The simulated spectra are plotted with some Lorentzian broadening. What is the value of the broadening parameter? This detail should be reported in the text.

Our calculations were carried out within harmonic approximation which does not allow one to estimate the width of spectral lines. All calculated Raman spectra presented in this paper were drawn using the Lorentzian line shape with ad hoc chosen with full width at half maximum of 8 cm-1 (unless otherwise specified). To avoid possible misunderstanding, the following sentences are added in section "Experimental and Calculation details":

“The phonon eigenvectors, frequencies, and Raman tensor components were calculated in the Г-point of the BZ within the density functional perturbation theory (DFPT) [23,24]. The Raman spectra were simulated using Lorentzian line shape functions with full width at half maximum of 8 cm-1 (unless otherwise specified)”.

2) In Figure 3, LA patterns are plotted. One should specify that these phonons are longitudinal indeed. By looking at the Figure, one can imagine that they are transverse.

We thank you for this remark, as it helped us to find out that the GaN and AlN layer designations in Figure 3 were incorrect. We have corrected this error, made different colors of the amplitudes of the displacements of the Ga and Al atoms, and indicated in the figure caption that the displacements of the atoms are directed along the c-axis.

3) In page 12, it is written that "This dependence has become the subject of numerous studies." The authors should cite these studies, at least the most important ones.

It is done. We have modified the sentence like given below.

This dependence has become the subject of numerous studies [45–49], the result of which has been the determination of the so-called deformation potentials that relate frequency shifts of the Raman lines with the homogeneous strain of the crystals.

4) It would be desirable to make a Figure showing all modes together, to get a clearer idea on the relative intensity of the different modes, and their relative position.

It is done. Figure S3 is presented in Supplementary Materials.

5) Typo: Bottom of page 10. The parenthesis in "geometry)" should be removed.

We have made this correction.

6) We also made some stylistic corrections in the text.

We hope that the revised version of the manuscript meets now the requirements necessary for its publication.

Sincerely,

Valery Davydov

Reviewer 2 Report

This is an excellent paper that systematically looks at phonon spectra of nitride superlattices from a theoretical and experimental perspective and using group theoretical methods gives insight into how to analyse the results. Aside from a thorough review of the written English, this is a very useful and well written paper.

Author Response

Dear Reviewer,

We are very grateful to you for careful reading of our manuscript and for useful suggestions on the paper grammar and style.

We completely agree with your proposal on improving English language and revised the manuscript in accordance with your recommendations.

Meeting your recommendations, English of the whole paper has been improved. In particular, we revised the correct use of definite and indefinite articles, prepositions and conjunctions. We have rewritten some of the sentences throughout the text to make them more clear and transparent.

We are very grateful to you and hope that the present version of the paper will meet all the requirements of the Journal.

Sincerely,

Valery Davydov

Reviewer 3 Report

This paper proposed a comprehensive group-theoretical analysis of short period AlN/GaN SLs and compared the Raman spectra of Ab initio calculation and the experiments.
The detailed analysis and clear explanations provide fundamental understanding of AlN/GaN SLs and will be useful to optimize the growth process. I recommend this paper can be published in Nanomaterials if following issues are well discussed.

1. Fig2. Fig4. Calculated Raman peaks are sharper than the experimental ones. Could authors comment on the broadening of Raman peaks in experiment?

2. Fig.3. The atomic displacement pattern of LA1 is symmetric in GaN and AlN regions. Because of different atoms, the amount of displacement can be different in GaN and AlN.
Is the symmetric pattern proper description?

3. Fig.4. Raman activity behaviors (peak heights) of TA and 2TA are considerably different between the calculated and experimental results.
Authors should discuss the height differences.

4. In this paper, lots of Raman spectra are compared for the calculated and experimental results. Of course, overall peak dependence on the period are well matched.
However, detailed spectral shapes - linewidth, peak positions, relative peak heights- are different. Could authors discuss the limit of the proposed analysis model?

Author Response

Dear Reviewer,

We are grateful to you for the attentive reading of our manuscript and valuable comments.

In accordance with your criticism, we have made the following corrections to the text of the manuscript:

1) Fig2. Fig4. Calculated Raman peaks are sharper than the experimental ones. Could authors comment on the broadening of Raman peaks in experiment?

Our calculations were carried out within harmonic approximation which does not allow one to estimate the width of spectral lines. All calculated Raman spectra presented in this paper were drawn using the Lorentzian line shape with ad hoc chosen with full width at half maximum of 8 cm-1 (unless otherwise specified). To avoid possible misunderstanding, the following sentences are added in section "Experimental and Calculation details":

“The phonon eigenvectors, frequencies, and Raman tensor components were calculated in the Г-point of the BZ within the density functional perturbation theory (DFPT) [23,24]. The Raman spectra were simulated using Lorentzian line shape functions with full width at half maximum of 8 cm-1 (unless otherwise specified)”.

2) Fig.3. The atomic displacement pattern of LA1 is symmetric in GaN and AlN regions. Because of different atoms, the amount of displacement can be different in GaN and AlN.

Is the symmetric pattern proper description?

Figure 3 represents eigenvectors of the folded LA-modes as they result from the diagonalization of the dynamic matrix calculated within the DFTP method. Indeed, there is a difference in atomic amplitudes in different layers. The difference is considerable for the optic modes (see Fig. 9 for example), but it is rather small for the acoustic modes. Acoustic modes even in a layered media correspond to homogeneous strains and are free of discontinuity in atomic displacements.

We thank you for this remark, as it helped us to find out that the GaN and AlN layer designations in Figure 3 were incorrect. We have corrected this error, made different colors of the amplitudes of the displacements of the Ga and Al atoms, and indicated in the figure caption that the displacements of the atoms are directed along the c-axis.

3) Fig.4. Raman activity behaviors (peak heights) of TA and 2TA are considerably different between the calculated and experimental results.

Raman peaks corresponding to TA phonons should be observed in the scattering geometry . Here z is the direction of the 3-fold optical axis. This means that for such measurements it is necessary to use a sample cleavage. Since the SLs samples were grown on a sapphire substrate, it is very difficult to obtain a high-quality, perfectly smooth cleavage. The presence of irregularities on the cleavage surface inevitably leads to depolarization of the measured spectrum, when strong peaks from the spectrum of a different scattering geometry can appear in it. Experimental Raman spectra for the (GaN)8/(AlN)8 SL observed in x(yz)x and x(yy)x scattering geometries are shown here in Fig.1. An analysis of the spectra revealed that the data, which were used for Fig. 4 of the manuscript (curve 2 in Fig 1a), include significant contributions of the spectrum corresponding to the x(yy)x geometry (curve 1 in Fig 1a).

This is especially evident in the region of optical phonons (see Fig. 1b), where the arrows show the peaks that should not be observed in the x(yz)x symmetry spectrum. In the region of acoustic phonons, the frequency of the 2TA peak in x(yz)x  symmetry spectrum almost coincides with the frequency of the 1LA peak which is observable in the x(yy)x scattering geometry. Therefore, it can be assumed that the increase in the intensity of the 2TA line is associated with the contribution of the 1LA mode, due to the significant depolarization of the spectrum.

In view of your comment, we have made additional efforts to improve the quality of polarization of the spectra measured from the cleavage of the sample. The best spectrum for SL 8/8 is shown by curve 3 in Fig.1a. Here the ratio of the intensities of the 1TA and 2TA peaks is close to the calculated one, and it is this spectrum that is inserted into the updated Fig.4 of the manuscript.

Figure 2 shows the similar x(yz)x and x(yy)x Raman spectra for SL 4/4. The analysis of the spectra leads to conclusion that the discrepancy in intensities of the 1TA and 2TA peaks in the calculations and experiment is also caused by the depolarization of the experimental spectrum. The best spectrum for SR 4/4, obtained after your comment, is shown by curve 3 in Fig. 2a. Here, the ratio of the intensities of the 1TA and 2TA peaks is close to the calculated one, and it is this spectrum that is inserted into the updated Fig.4 in the manuscript.

We are extremely grateful to you for such a careful reading of our manuscript and for the found contradictions between theory and experiment, the elimination of which, undoubtedly, improved the quality of the material presented.

4) In this paper, lots of Raman spectra are compared for the calculated and experimental results. Of course, overall peak dependence on the period are well matched.

However, detailed spectral shapes - linewidth, peak positions, relative peak heights- are different. Could authors discuss the limit of the proposed analysis model?

Comparison of calculated and experimental Raman spectra throughout of the paper reveals good agreement with respect to peak position (i.e. the phonon frequencies). The disagreement is lower than 10% which is a generally recognized threshold for the modern non-empiric DFT calculations (e.g. see. Petretto, G., et al. Sci Data 5, 180065 (2018). doi: 10.1038/sdata.2018.65).  Agreement with respect to the Raman intensity is worse. Nevertheless, it is sufficient to confirm the peak assignment made basing on the peak position. The linewidths were not discussed in the paper. Emphasize that the coincidence of calculated and experimental Raman spectra was not the main issue of the paper. The achieved agreement allowed us to confirm a good quality of the samples under study (note that the calculations were based under assumption of perfect interfaces) and to show how much of valuable structural information can be drawn from Raman spectra of  the short-period SLs.

5) We also made some stylistic corrections in the text.

We hope that the revised version of the manuscript meets the requirements necessary for its publication.

Sincerely,

Valery Davydov
